# Single Cell Transcriptome Analysis of Niemann–Pick Disease, Type C1 Cerebella

**DOI:** 10.3390/ijms21155368

**Published:** 2020-07-28

**Authors:** Antony Cougnoux, Julia C. Yerger, Mason Fellmeth, Jenny Serra-Vinardell, Kyle Martin, Fatemeh Navid, James Iben, Christopher A. Wassif, Niamh X. Cawley, Forbes D. Porter

**Affiliations:** 1Division of Translational Medicine, *Eunice Kennedy Shriver* National Institute of Child Health and Human Development, National Institutes of Health, Bethesda, MD 20892, USA; antony.cougnoux@gmail.com (A.C.); julia.yerger@nih.gov (J.C.Y.); fellm033@gmail.com (M.F.); ky.martn@gmail.com (K.M.); wassifc@cc1.nichd.nih.gov (C.A.W.); cawleyn@mail.nih.gov (N.X.C.); 2Human Biochemical Genetics Section, National Human Genome Research Institute, National Institutes of Health, Bethesda, MD 20892, USA; jenny.serra-vinardell@nih.gov; 3Pediatric Translational Research Branch, National Institute of Arthritis and Musculoskeletal and Skin Disease, National Institutes of Health, Bethesda, MD 20892, USA; fatemeh.navid@nih.gov; 4Molecular Genomic Core, *Eunice Kennedy Shriver* National Institute of Child Health and Human Development, National Institutes of Health, Bethesda, MD 20892, USA; james.iben@nih.gov; 510CRC, Rm. 5-2571, 10 Center Dr., Bethesda, MD 20892, USA

**Keywords:** Niemann–Pick disease, type C1, NPC1, single cell RNA sequencing, transcriptome, cerebellum, cerebellar ataxia

## Abstract

Niemann–Pick disease, type C1 (NPC1) is a lysosomal disease characterized by endolysosomal storage of unesterified cholesterol and decreased cellular cholesterol bioavailability. A cardinal symptom of NPC1 is cerebellar ataxia due to Purkinje neuron loss. To gain an understanding of the cerebellar neuropathology we obtained single cell transcriptome data from control (*Npc1^+/+^*) and both three-week-old presymptomatic and seven-week-old symptomatic mutant (*Npc1^−/−^*) mice. In seven-week-old *Npc1^−/−^* mice, differential expression data was obtained for neuronal, glial, vascular, and myeloid cells. As anticipated, we observed microglial activation and increased expression of innate immunity genes. We also observed increased expression of innate immunity genes by other cerebellar cell types, including Purkinje neurons. Whereas neuroinflammation mediated by microglia may have both neuroprotective and neurotoxic components, the contribution of increased expression of these genes by non-immune cells to NPC1 pathology is not known. It is possible that dysregulated expression of innate immunity genes by non-immune cells is neurotoxic. We did not anticipate a general lack of transcriptomic changes in cells other than microglia from presymptomatic three-week-old *Npc1^−/−^* mice. This observation suggests that microglia activation precedes neuronal dysfunction. The data presented in this paper will be useful for generating testable hypotheses related to disease progression and Purkinje neurons loss as well as providing insight into potential novel therapeutic interventions.

## 1. Introduction

Niemann–Pick disease, type C1 (NPC1, MIM #257220) is a rare neurodegenerative disease caused by reduced function of NPC1 [1]. The NPC1 protein, in conjunction with NPC2, functions to transport cholesterol out of the endolysosomal compartment [2]. Decreased NPC1 or NPC2 function leads to endolysosomal storage of unesterified cholesterol and other lipids [3,4]. Mutations of *NPC1* account for approximately 95% of the cases of NPC, and the other 5% are due to pathogenic variants in *NPC2* [5]. Data from large sequence databases are consistent with an incidence of NPC1 on the order of 1/90,000 and suggest that there may be a late-onset NPC1 phenotype with a significantly higher incidence [6]. During the neonatal period, infants with NPC1 may present with cholestatic liver disease [7], but after the neonatal period, progressive neurological disease dominates the clinical picture. Characteristic neurological manifestations include progressive supranuclear gaze palsy, gelastic cataplexy, seizures, cognitive impairment, and cerebellar ataxia [5,8,9].

Cerebellar ataxia is a cardinal symptom of NPC1. The cerebellum accounts for more than half of the total number of neurons in the central nervous system (CNS) [10]. Its primary function is to coordinate motor control and coordination, but recent work suggests it also plays a role in other processes such as cognition [11]. The cerebellar cortex has a relatively simple three-layer organization [12]. The central layer is composed of a single layer of Purkinje neurons. Purkinje neurons are large inhibitory GABAergic neurons that function to integrate cerebellar neuronal input and provide the sole output of the cerebellum via axons that project to the deep cerebellar nuclei. The Purkinje neuron layer lies between the inner granule layer composed primarily of excitatory granule neurons, and the outer molecular layer composed primarily of granule neuron axons (parallel fibers) and the Purkinje neuron dendritic tree. In addition to the glutamatergic granule neurons, the granule layer also contains other neuronal subtypes including various interneurons such as inhibitory Golgi cells and glutamatergic unipolar brush cells, the latter of which function to amplify signals from the vestibular ganglia and provide information on spatial orientation. Basket cells, found in the molecular layer, synapse on the Purkinje neuron cell and provide inhibitory input. In addition to neurons, the cerebellum contains numbers of supporting glial cells (astrocytes, ependymal cells, and oligodendrocytes), vascular associated cells, and myeloid (microglia and monocytes/macrophages).

Cerebellar ataxia in NPC1 results from the progressive loss of cerebellar Purkinje neurons. Purkinje neuron loss in NPC1 occurs in a stereotypic anterior to posterior gradient with relative preservation of a subset of aldolase C positive Purkinje neurons [13]. Although Purkinje neuron loss has been reported to be cell autonomous [14], histopathological changes are observed in astrocytes and oligodendrocytes [15], and microglial activation is a predominant aspect of and likely contributor to NPC1 neuropathology [16]. *Npc1* expression in astrocytes significantly increases survival for *Npc1*^−/−^ mice, suggesting glial cells dysfunction is involved in disease progression [17]. Differences in RNA expression between control and mutant tissue can be used to obtain insight into biological processes that are altered in the disease state, and groups have compared cerebellar transcriptomes between control and NPC1 mutant mice at different ages [18,19,20,21]. However, transcriptome analysis of intact tissue/lobules mask potentially significant expression changes at the individual cellular level. Single cell transcriptome analysis has been extensively applied to characterize neuronal cell populations from the entire brain [22,23] and notably the cerebellum [24]. To gain insight into pathological processes occurring in individual cerebellar cell types we performed single cell RNA sequencing (scRNAseq) on cerebellar tissue collected from asymptomatic three-week-old and symptomatic seven-week-old male *Npc1^−/−^* (BALB/*Npc1*^m1N^) and corresponding control *Npc1^+/+^* littermates. Understanding the individual cellular contributions to NPC1 pathology may lead to therapeutic approaches targeting various aspects of the pathological cascade.

## 2. Results

### 2.1. Cell Type Specific Transcriptomes from Symptomatic 7-Week Old NPC1 Mice

Single cell RNA sequencing was used to obtain cerebellar single cell transcriptome data from seven-week-old male NPC1 mutant (*Npc1^−/−^*) and littermate control (*Npc1^+/+^*) mice. Two cerebella from each genotype yielded 1875 and 1430 single-cell transcriptomes from *Npc1^+/+^* and *Npc1^−/−^* tissue, respectively. Visualization by t-distributed stochastic neighbor embedding (t-SNE) allowed for the identification of clusters of cells with similar transcriptomes (Figure 1A and Appendix A). Cell-type-specific transcripts (signature transcripts) were used to identify the type of cell in the t-SNE clusters, and the number of cells identified for both genotypes is shown in Figure 1B. Based on expression of signature transcripts, we identified transcriptomes corresponding to myeloid cells (monocytes and microglia), vascular cells (endothelial, vascular smooth muscle, vascular leptomeningeal and arachnoid barrier), glial cells (astrocytes, oligodendrocytes, ependymal) and neurons (basket, unipolar brush, cerebellar granule, interneurons, Purkinje). Appendix A shows the distribution of specific signature transcripts corresponding to cell transcriptome clusters on the t-SNE plot. The signature transcripts used to identify most cell types correspond to known histological markers (Figure 1C) and have been used by others to identify cell type specific transcriptomes in scRNAseq experiments [22,23,25]. In the cerebellum calbindin, there is an immunohistochemical marker of Purkinje neurons, IBA1 and CD68 label microglia, TBR2 (product of *Eomes*) labels unipolar brush neurons, CD117 (product of *Kit*) labels basket/stellate neurons, OLIG2 labels oligodendrocytes, NEUROD1 labels cerebellar granule neurons, GFAP and AQP4 are astrocyte markers, NEUN (product of *Snap25*) labels most neurons, and parvalbumin is used to identify interneurons. Based on transcriptome data (Appendix A) and immunohistochemical staining (Figure 1C), we show that aquaporin 1 (*Aqp1*) can be used to identify ependymal secretory cells. From the scRNAseq data, there appeared to be a decreased number of astrocytes, oligodendrocytes, granule neurons and unipolar brush neurons in the *Npc1* mutant cerebella when compared to the control data (Figure 1B). However, these apparent differences could not be confirmed experimentally (Figure 1D). This discrepancy may be due to increased sensitivity of *Npc1^−/−^* cells to the enzymatic disassociation required to prepare samples for scRNAseq. In the following sections we will catalogue and compare cerebellar cell type specific transcriptome data from seven-week-old *Npc1^+/+^* and *Npc1^−/−^* mice.

### 2.2. Myeloid Transcriptomes

Consistent with previous work on immune cells in the brain of *Npc1^−/−^* mice [16,26], we did not observe infiltrating peripheral immune cells other than perivascular monocytes/macrophages. Monocytes and microglia are both characterized by the high expression of *C1qa*, while *Pf4* (also named *Cxcl4*) and *Tmem119* are respectively specific for monocytes and microglia (Appendix A, [25]). No significant gene expression differences were observed comparing *Npc1* control and null monocytes. However, consistent with our prior work characterizing transcript levels in FACS isolated microglia [27], we observed major gene expression differences between *Npc1^+/+^* and *Npc1^−/−^* microglia.

Microglia are central nervous system resident immune cells that originate from the yolk sac during development [28]. The t-SNE plot identified two populations of microglia (Figure 1A and Appendix A). Principal component analysis (PCA) of the microglia transcriptomes shows a distribution of the cells along the first dimension of the PCA based on genotype (Figure 2A). *Npc1*^−/−^ microglia can be differentiated from *Npc1^+/+^* microglia based on reduced expression of the linage markers *Csf1r*, *Cst3*, *Cx3cr1*, *P2ry13* and *Tmem119* (Figure 2B, [27]). A continuum of activation between the non-activated ramified state and the reactive amoeboid state is suggested by increased expression of *Cd22*, *Cd68*, *Gapdh*, *Hpse*, *Igf1*, *Itgax*, and *Trem2* in *Npc1*^−/−^ microglia (Figure 2B,C). This expression pattern is consistent with a subset of the microglia in *Npc1^−/−^* being classified as disease associate microglia (DAM). DAM were initially described in an Alzheimer disease mouse model and are thought to be neuroprotective [29], thus supporting the concept that subpopulations of microglia in NPC1 have neurotoxic or neuroprotective activity [27]. Decreased expression of *Csf1r* in *Npc1^−/−^* microglia likely explains a prior unpublished observation that treatment of *Npc1^−/−^* mice with PLX3397, a CSF1R inhibitor used to deplete microglia, did not have a demonstratable effect on either cerebellar histopathology or survival of *Npc1^−/−^* mice (Appendix A).

Pathway analysis of differentially expressed genes showed similarities in microglia activation between NPC1 and common neurodegenerative diseases (Appendix A, [30]). Thus, we compared our results with single cell transcriptome data from amyotrophic lateral sclerosis and Alzheimer disease mouse models [29]. Based on transcriptome analysis, activated microglia isolated from seven-week-old *Npc1*^−/−^ cerebellum more closely resemble those from the Alzheimer disease mouse model (r_Pearson_ = 0.55, *p* < 0.001) than the amyotrophic lateral sclerosis mouse model (r_Pearson_ = −0.48, *p* < 0.001) when compared to their respective controls (Appendix A). Expression of neuroprotective genes *Igf1*, *Gpnmb* and *Spp1* is increased in microglia from both NPC1 and Alzheimer disease mouse models. In contrast, expression of these three genes is decreased in microglia from the amyotrophic lateral sclerosis mouse model. Flow cytometry analysis of cerebellar microglia from control and *Npc1* mutant mice confirmed increase protein expression of DAM markers CD11c (product of *Itgax*) and apolipoprotein E (*Apoe*) in *Npc1^−/−^* cerebellar microglia (Appendix A).

### 2.3. Vascular Associated Cell Transcriptomes

Single cell sequencing can identify multiple cell types associated with central nervous system blood vessels [23]. In our data set, we were able to identify four vascular associated cell types. These included vascular smooth muscle, endothelial cells, arachnoid barrier cells, and vascular leptomeningeal cells (Figure 1A,B and Appendix A). Vascular smooth muscle cells are characterized by the expression of *Acta2*, *Myh11*, *Myl9*, and *Tagln* [25]. Data for *Acta2* are shown in Appendix A. Appendix A provides a description of differentially expressed genes and their relevance to NPC1 for all identified cell types. When looking at genes with increased expression, the most enriched pathway is “vascular smooth muscle cell contraction,” which supports our identification (Appendix A). Comparison of *Npc1^+/+^* and *Npc1^−/−^* vascular smooth muscle gene expression showed four genes with a significant differential expression: *Apoe*, *Ctsd*, *Lyz2*, and *Map3k7cl* (Figure 3A). Endothelial cells are characterized by expression of *Cldn5* and *Pecam1* (Figure 1 and Appendix A). Pathway analysis of genes enriched in endothelial cells support the exchange function of these cell between the circulation and tissue and nitric oxide signaling important for angiogenesis (Appendix A). Comparison between genotypes identified five genes with significant differential expression in *Npc1*^−/−^ endothelial cells. The differentially expressed genes are *Apoe*, *Ctla2a*, *Ctsb*, *Ctsd*, and *Lyz2* (Figure 3B, Appendix A). Vascular leptomeningeal cells and arachnoid barrier cells are identified by high expression levels of *Dcn*, *Col1a1* and *Col1a2* (Figure 1 and Appendix A). They are differentiated by expression of *Colec12* by vascular leptomeningeal cells and *Slc47a1* by arachnoid barrier cells (Appendix A). No differentially expressed genes were identified in arachnoid barrier cells. The low number of transcriptomes corresponding to arachnoid barrier cells likely decreased our sensitivity to identify differentially expressed genes. Similar to vascular smooth muscle and endothelial cells, increased expression of *Ctsd*, *Lyz2*, and *Tyrobp* was observed in *Npc1^−/−^* vascular leptomeningeal cells (Figure 3C). Pathway analysis on genes enriched in both arachnoid barrier cells and vascular leptomeningeal cells support the similarities between these cells and their involvement in vasculature. Insulin-like growth factor is involved in blood vessel homeostasis [31], and heparin binding is enriched in both clusters (Appendix A).

### 2.4. Glial Cell Transcriptomes

Glial cells in the cerebellum include astrocytes, oligodendrocyte precursors, oligodendrocytes, and, due to its proximity with the fourth ventricle, both ciliated and secretory ependymal cells. Astrocytes are characterized by the expression of *Slc1a3* (Appendix A)*. Slc1a3* encodes SLC1A3 (GLAST, EAAT1), the main ionotropic glutamate transporter of the cerebellum [32]. Pathway analysis on genes enriched in astrocytes highlight the supportive functions of these cells in buffering ions and neurotransmitters (Appendix A). *Npc1^+/+^* astrocytes partition into two distinct populations (Appendix A) corresponding to “homeostatic” and “reactive” astrocytes [33]. Reactive astrocytes are characterized by decreased expression of both *Slc1a3* and *Gdf10* (growth differentiation factor 10 or bone morphogenetic protein 3B) and increased expression of both *Aqp4* (Aquaporin-4) and *Gfap* (Glial fibrillary acidic protein) (Appendix A). In contrast, *Npc1^−/−^* astrocytes are not clearly separated into the homeostatic and reactive populations (Appendix A). Instead *Npc1^−/−^* astrocytes have transcriptomes that represent a continuum between the homeostatic and reactive states (Appendix A). Unfortunately, the depth of the sequencing did not allow us to determine the “polarity” of the *Npc1^−/−^* reactive astrocytes (*Gfap*^high^*Aqp4*^high^) based on polarization markers suggested by Liddelow et al. [33]. Activated microglia induce A1 astrocyte polarization (neurotoxic) by secretion of IL-1α, TNF, and C1q. However, the absence of a significant phenotypic difference between *Npc1*^−/−^:*C1qa*^+/+^ and *Npc1*^−/−^:*C1qa*^−/−^ mice [19] suggests that *Npc1^−/−^* astrocytes do not acquire an A1 neurotoxic polarity. Nine transcripts corresponding to *Agt*, *Aqp4*, *Ctsb*, *Ctsd*, *Gfap*, *Lyz2*, *Mt1*, *Mt2*, and *Ttr* were observed to have significantly increased expression in *Npc1^−/−^* astrocytes relative to littermate control mice (Figure 4A, Appendix A).

Transcriptomes corresponding to oligodendrocytes are identified based upon high expression of *Mo**g*, *Olig1*, *Olig2*, and *Mag*. Data corresponding to *Mog* is shown in Appendix A. T-SNE plots of oligodendrocytes, independent of genotype, show two clusters (Appendix A). These two clusters correspond to precursor (*Marcks* and *Gpr17* positive) and differentiated (*Mog* and *Car2* positive) oligodendrocytes (Appendix A). Supporting this identification, pathways with increased expression include axon ensheathment, oligodendrocytes development and myelination (Appendix A). Eleven genes with significantly increased expression were identified in *Npc1^−/−^* differentiated oligodendrocytes. These were *Apoe*, *B2m*, *Cd63*, *Ctsb*, *Ctsd*, *Fabp5*, *Gstp1*, *Lyz2*, *Npc2*, *Ptgds*, and *S100a1* (Figure 4B, Appendix A). Five transcripts were significantly overexpressed in *Npc1^−/−^* precursor oligodendrocytes. *Apoe*, *Ctsd*, and *Lyz2* were overexpressed in both precursor and differentiated oligodendrocytes, whereas increased expression of *Arhgap31* and *Tyrobp* was only observed in precursor oligodendrocyte transcriptomes (Figure 4C, Appendix A).

Ependymal cells are organized in a tight monolayer that lines the ventricles of the brain forming a physical barrier between brain tissue and the cerebral spinal fluid (CSF) filled ventricles (reviewed in [34]). Ependymal cells are involved in the production and regulation of the composition of cerebrospinal fluid. Via the action of luminal cilia, ependymal cells promote circulation of CSF. In the cerebellum they form the epithelial layer of the fourth ventricle. Whereas the function of specific proteins involved in forming a tight junction barrier between the ventricles and the brain (ZO1, OCLN, and CX43) and hydrocephalic pressure regulation (Aquaporin 4 and 11) have been studied, relatively little is known regarding the biology of these cells [35]. There are two main types of ependymal cells, ciliated and secretory. Ciliated ependymal cells are characterized by high expression of *Ccdc153* and *Tmem212* and secretory ependymal cells are characterized by high expression of *Aqp1* and *Sostdc1* (Appendix A). Analysis of ependymal ciliated cell transcriptomes confirmed the expression of cilium related genes and the lack of angiogenesis or stem cell related function as described by Shah et al. [36] (Appendix A). Comparison of *Npc1^+/+^* and *Npc1^−/−^* transcriptomes showed increased expression of *C1qa*, *Lyz2*, and *Tyrobp* (Figure 1D). Consistent with their hydrodynamic and metabolic function, the most highly enriched biological processes in ependymal cells were primary energy metabolism and transmembrane transport (Appendix A). Comparison of *Npc1^+/+^* and *Npc1^−/−^* transcriptomes showed significantly increased expression of 16 genes and significantly decreased expression of one gene in *Npc1^−/−^* secretory ependymal cells (Figure 4E, Appendix A). The genes showing increased expression were *Apoe*, *B2m*, *C1qa*, *C1qb*, *C1qc*, *Ctsl*, *Ctss*, *Ctsz*, *Fabp5*, *Fcer1g*, *Hexb*, *Lyz2*, *Spp1* (Osteopontin), *Tmsb4x*, *Trem2*, and *Tyrobp*. The only gene with significantly decreased expression was *Manf.*

### 2.5. Neuronal Transcriptomes

The majority of neurons can be identified by expression of synaptosome-associated protein 25 (*Snap25*) encoding the protein NEUN (Figure 1C and Appendix A). Among *Snap25*-positive cells, specific neurons can be differentiated based on expression of characteristic transcripts. This includes expression of Eomesodermin (*Eomes*/TBR2) in unipolar brush cells, calbindin 1 (*Calb1*) and ryanodine receptor 1 (*Ryr1*) in Purkinje neurons, cerebellin-3 precursor (*Cbln3*) and neurogenic differentiation 1 (*Neurod1*) in cerebellar granule cells, and CD117 (*Kit*) in basket/stellate cells (Figure 1C and Appendix A).

Basket and stellate cells are GABAergic inhibitory interneurons present in the cerebellar molecular layer identified by expression of *Snap25* and *Kit* (coding for NEUN and CD117, respectively, Appendix A). Although basket and stellate neurons are morphologically distinct and innervate Purkinje neurons differently [37], we were not able to differentiate their transcriptomes. Pathway analysis of basket/stellate cell transcriptomes was notable for expression of genes encoding synaptic GABA receptors, neurotransmitter transporters and ion transporters (Appendix A). A significant increase in expression of *Apoe*, *Ctsd*, *Lyz2*, and *Ttr* was observed in *Npc1^−/−^* basket/stellate neurons (Figure 5A).

Cerebellar granule neurons are glutamatergic neurons localized in the inner granule layer. They are the most common neuron in the cerebellum. Cerebellar granule neurons receive input from mossy fibers coming from the pontine nuclei and project axons into the molecular layer, where they form excitatory synapses on Purkinje neuron dendrites. Transcriptomes corresponding to cerebellar granule neurons were identified by expression of *Snap25*, *Neurod1*, and *Clbn3* (Appendix A) and showed enrichment of genes involved in glutamatergic signaling (Appendix A). Like what we observed for basket/stellate neurons, the cerebellar granule neurons from *Npc1* mutant mice have increased expression of *Apoe* and *Ctsd* (Figure 5B).

Unipolar brush cells are small, *Eomes*/TBR2 positive (Figure 1 and Appendix A) glutamatergic neurons, residing in the granular layer of the cerebellar cortex [38]. In *Asic5* mutant mice, dysfunction of unipolar brush cells results in cerebellar ataxia [39]. Pathway analysis confirmed the glutamatergic signaling of these cells (Appendix A). Four transcripts show significantly increased expression in *Npc1*^−/−^ unipolar brush cells (Figure 5C). Like basket/stellate and cerebellar granule neurons, unipolar brush cells have increased expression of *Apoe* and *Ctsd*. Expression of both *Lyz2* and *Tyrobp* was also increased.

Purkinje neurons are very large neurons organized in a monolayer between the molecular and granular layer of the cerebellar cortex (Figure 1 and Appendix A). Purkinje neurons are GABAergic neurons that integrate inputs from other cerebellar neurons and project axons to the deep cerebellar nuclei. Enriched pathways are consistent with GABAergic signaling (Appendix A). These neurons are of particular interest because progressive rostral to caudal loss of Purkinje neurons is a pathological hallmark of NPC1 [40], and loss of Purkinje neurons results in cerebellar ataxia, a major clinical feature of NPC1 [5,8,9]. Purkinje neurons transcriptomes were identified by high expression of *Calb1*, *Car8*, and *Ryr1* [22] (Appendix A).

Clustering analysis identified two distinct populations of Purkinje neurons, but did not separate based on genotype (Appendix A). One of the clusters consisted of transcriptomes from 12 *Npc1^+/+^* Purkinje neurons and the other cluster included 22 transcriptomes evenly distributed between *Npc1^+/+^* and *Npc1^−/−^* Purkinje neurons. We suspected that the clustering might be related to location of the Purkinje neurons in the cerebellum, since Purkinje neurons in anterior lobules (I to VI) are substantially lost by seven weeks of age. Classification of these two clusters as anterior and posterior Purkinje neurons (Appendix A) is supported by prior data and the Allen Brain Atlas [20,41]. *Calb1* and *Car8*, known Purkinje neuron markers, show uniform expression, whereas *Sv2c* and *Abdh3* show decreased and *Car7* and *B3galt5* [42] show increased expression in posterior Purkinje neurons (Appendix A). This is consistent with in situ expression data from the Alan Brain Atlas (Appendix A). This assignment was supported by looking at expression of aldolase C (*Aldoc*). Aldolase C protein is expressed in Purkinje neurons that are relatively resistant to neurodegeneration in NPC1 [13], and *Aldoc* expression is increased in posterior lobules (https://porterlab.shinyapps.io/cerebellarlobules/). Mean *Aldoc* expression was decreased in *Npc1^+/+^* Purkinje neurons in the anterior cluster (4777 ± 3282 CPM) versus those from the posterior cluster (14,692 ± 5325 CPM). Mean *Aldoc* expression (12,130 ± 9769 CPM) in the *Npc1^−/−^* neurons was similar to the posterior cluster *Npc1^+/+^* Purkinje neurons.

Differential gene expression analysis between *Npc1*^−/−^ and posterior *Npc1*^+/+^ Purkinje neurons showed 11 genes with a differential expression: *Apoe*, *B2m*, *C1qa*, *Ctsd*, *Ctss*, *Ctsz*, *Fstl4*, *Fxyd7*, *Lyz2*, *Them6*, and *Tyrobp* (Figure 5D, Appendix A). Increased expression of *Apoe*, *Ctsd*, *Lyz2 and Tyrobp* in *Npc1^−/−^* Purkinje neurons overlap with what we observed in other cerebellar neurons (Figure 5A–C), and increased expression of *B2m*, *C1qa*, *Ctss*, *Ctsz*, and *Tyrobp* was observed in other *Npc1^−/−^* cerebellar cell type transcriptomes.

### 2.6. Protein Validation of Selective Genes Demonstrating Increased Expression in Npc1^−/−^ Transcriptomes

Due to the multiple factors that influence translation and protein stability, proteomic and transcriptomic results do not always correlate [43]. Therefore, we evaluated protein expression of cathepsin D (*Ctsd*), lysozyme (*Lyz2*), DAP12 (*Tyrobp*), and APOE (*Apoe*) in cerebellar tissue from *Npc1^+/+^* and *Npc1^−/−^* mice. Expression of β2 microglobulin (*β2m*) was quantified using an Enzyme Linked Immunosorbent Assay (ELISA). Consistent with the transcriptomic results, increased protein expression in *Npc1^−/−^* cerebellar tissue was confirmed for cathepsin D, lysozyme, DAP12, and β2 microglobulin (Figure 6 and Appendix A). In contrast, APOE levels were not elevated. Both lysozyme and cathepsin D have been shown to be increased in NPC1 patient serum [44,45]; thus, they, along with other secreted proteins with elevate expression may be potential biomarkers of disease status in cerebral spinal fluid. Increased protein expression could indicate functional relevance to NPC1 pathology that can be explored in future studies.

### 2.7. Cell Type Specific Transcriptomes from Asymptomatic Three-Week-Old NPC1 Mice

To explore early changes taking place in the cerebellum of *Npc1^−/−^* mice, we obtained scRNAseq data from asymptomatic 3-week-old animals. At this age there is significant unesterified cholesterol storage, but neither gliosis nor Purkinje neurons loss has yet occurred [16,46]. We obtained single-cell transcriptome data from 2738 *Npc1*^+/+^ and 1752 *Npc1*^−/−^ cells. At 3-weeks of age tSNE clustering identified transcriptomes corresponding to the same cerebellar cell types that we identified at 7-weeks of age (Appendix A). Comparison of *Npc1^+/+^* and *Npc1^−/−^* transcriptomes at 3-weeks of age showed no significant genotype differences for arachnoid barrier cells, vascular smooth muscle cells, vascular leptomeningeal cells, oligodendrocytes, ependymal cell, cerebellar granule neurons, stellate/basket neurons, unipolar brush neurons or Purkinje neurons. This surprising observation was not due to a low number of transcriptomes since at three weeks we had data for more individual transcriptomes for both genotypes than we had for the seven-week-old symptomatic mice. Significant differences were only observed for endothelial cells and microglia. In endothelial cells, we observed altered expression of four genes. These included increased expression of *Apoe*, *Timp3*, and *Mt2* and decreased expression of *Slc16a1* (Figure 7A, Appendix A). Only *Apoe* also showed increased expression in *Npc1^−/−^* endothelial cells at both three and seven weeks of age. Differential expression of either *Timp3* or *Slc16a1* was not observed in other cell types at either three or seven weeks.

## 3. Discussion

Comparison of transcriptome data between control and disease tissue can be used to obtain insight into biological processes. Although multiple groups, including ours, have studied whole tissue transcriptome data comparing NPC1 control and mutant cerebellar expression, to our knowledge this is the first report of single cell transcriptomic data from an NPC1 model system. Although scRNAseq is limited by the depth of coverage one can obtain, it has the potential to identify differential gene expression in individual cell types that could be masked when performing whole tissue RNAseq. In this paper we compare cerebellar scRNAseq data from *Npc1^+/+^* and *Npc1^−/−^* mice at both seven weeks (symptomatic) and three weeks (asymptomatic) of age. We focused on the cerebellum given that Purkinje neuron loss is a major pathological finding in NPC1, and the resulting cerebellar ataxia is a cardinal symptom in this disorder. The cell type specific data on differentially expressed genes and the subcellular location of the encoded protein are summarized in Figure 8. This dataset should be useful in further understanding the components of the pathological cascade that leads to Purkinje neuron loss in NPC1 disease.

We did not anticipate either the limited number of differentially expressed transcripts or that the same set of transcripts (namely *Apoe*, *Ctsd*, *Lyz2*, and *Tyrobp*) would be identified in multiple different cell types in the seven-week data. The low number of differentially expressed genes identified might be due to limited sequencing depth inherent to scRNAseq. Arguing against this explanation is that we were able to, as expected from prior work, identity a large number of differentially expressed genes in microglia. Alternatively, the small number of genes with altered expression could indicate that cerebellar pathology is secondary to pathology in another brain region. The cerebellum is highly connected to the thalamus via the cerebellothalamic tract, and the thalamus is one of the earliest regions demonstrating neuropathology in the *Npc1^−/−^* mouse model [47]. Similarly, Gurda et al. [48] showed that autophagic defects in climbing fibers originating from the inferior olivary nuclei correlate both temporally and specially with Purkinje neuron loss in the NPC1 cat model. The concept that cerebellar pathology might be driven by pathology in another brain may explain how limited Purkinje neuron transduction by gene therapy results in significant Purkinje neuron sparing and survival [49].

Of the 38 genes identified as dysregulated in non-myeloid cells, 26% correspond to lysosomal proteins (Figure 8). This was anticipated based on prior work describing increased expression of lysosomal proteins in NPC1 models [50,51,52]. Of the 36 genes showing increased expression, 18 (50%) encode secreted proteins. Promoting lysosomal exocytosis has been proposed as a mechanism for clearance of stored cholesterol [53,54,55,56]. Thus, it is possible that this represents minor pathway that cells utilize to redistribute cholesterol resulting in increased expression of secreted proteins to compensate for their loss. Secreted proteins are also candidates for biomarkers [45]. Interestingly, 42% (16/38) of the dysregulated genes encode sphingolipid-binding proteins [57] (Figure 8). Sphingosine and sphingolipids accumulate in NPC1 and have been proposed to contribute to NPC pathology [58,59,60], and miglustat, a glycosphingolipid synthesis inhibitor, has shown efficacy in slowing NPC1 disease progression [61,62]. In contrast to the high fraction of sphingolipid-binding proteins, only three of the dysregulated genes (*Apoe*, *Ctsd*, and *Cd63*) encode proteins were shown by Hulce et al. [63] to interact with cholesterol. NPC2 is also known to bind cholesterol [64]. Determining whether this intersection between dysregulated gene expression and sphingolipid-binding proteins is of functional relevance will require future investigation.

As noted above, expression of four transcripts (*ApoE*, *Ctsd*, *Lyz2*, and *Tyrobp*) was elevated in multiple cerebellar cell types, including neurons, from *Npc1^−/−^* mice. *ApoE* encodes Apolipoprotein E (APOE). APOE is a major protein component of high-density lipoproteins that transport cholesterol between cells in the central nervous system. The APOE4 variant of APOE has been identified as a risk factor for Alzheimer’s disease [65] and is associated with increased disease severity in NPC1 patients [66]. Injured or stressed neurons increase synthesis of APOE [67], and this is consistent with our current observation of increased *Apoe* expression in basket/stellate, granule, unipolar and Purkinje neurons in *Npc1^−/−^* cerebella. *Ctsd* encodes cathepsin D, a lysosomal aspartyl protease. Increased expression of cathepsin D has been reported NPC1cerebellum and it has been proposed that increased cathepsin D enzyme activity could contribute to neuronal vulnerability in NPC1 [68]. *Lyz2* encodes lysozyme, a glycoside hydrolase that functions in the innate immunity response against gram-positive bacteria. Increased expression of lysozyme has previously been observed in NPC1 [18,19,20,21,44,69]; however, this has been assumed to be secondary to immune cell activation. Our current data show that *Lyz2* is highly expressed by multiple non-myeloid cell types including basket/stellate, unipolar brush, and Purkinje neurons. It is possible that elevated expression of lysozyme in Purkinje neurons functionally contributes to their loss. Prior work on Sanfilippo syndrome type B has shown although lysozyme expression is ubiquitous, lysozyme protein accumulates specifically in disease susceptible neurons, and this group correlated lysozyme protein expression with formation of hyperphosphorylated tau in the same neurons [70]. Alterations in tau function have previously been proposed as a component of the NPC1 pathology [71,72,73,74,75]. *Tyrobp* encodes TYRO protein tyrosine kinase binding protein (DAP12). DAP12 functions as a signaling adaptor protein for multiple plasma membrane receptors. DAP12 appears to have multiple functions including involvement in bone modeling, myelination and inflammation [76]. It is likely that TYROBP plays a role in neuroinflammation in that loss of DAP12 function delays disease progression in a mouse Alzheimer disease model [77]. However, TYROBP genetic variants may predispose to early onset Alzheimer disease [78]. Further work will be needed to determine if increased expression of these four genes is involved in the cerebellar pathology of NPC1 or if they can be utilized as biomarkers to monitor neurological disease.

Purkinje neuron loss and resulting cerebellar ataxia are cardinal features of NPC1, thus we were particularly interested in determining transcriptomic changes in NPC1 mutant Purkinje neurons. Our seven-week scRNAseq data was sensitive enough to separate anterior and posterior Purkinje neurons. In the surviving posterior *Npc1^−/−^* Purkinje neurons, we identified 11 genes with significantly altered expression. These included the four transcripts (*ApoE*, *Ctsd*, *Lyz2*, and *Tyrobp*) discussed above, two additional cathepsins (*Ctss* and *Ctsz*), *B2m*, *C1qa*, *Fstl4*, *Fxyd7*, and *Them6*. CTSS but not CTSZ protein expression has been shown to be increased in lysosomes isolated from NPC1 mutant mouse brain tissue [79]. Dysregulation of cathepsin enzymatic activity has been proposed to be a pathogenic factor in multiple lysosomal storage diseases including NPC1 [80,81,82]. *B2m* encodes Beta2-microglobulin and increased CSF levels have been associated with neuroinflammation [83]. *C1qa* encodes a subunit of complement C1. Increased expression of *C1qa* does not appear to contribute to NPC1 neuropathology since neurological disease progression is not altered in *Npc1^−/−^*:*C1qa^−/−^* double mutant mice [19]. *Fxyd7* encodes a protein, FXYD7, which interacts with NaK-ATPase to regulate neuronal excitability [84]. *Fstl4* encodes follistatin-related protein 4 precursor (SPIG1), a protein that can bind pro-BDNF (Brain Derived Neurotrophic Factor) and impairs it maturation [15]. It has been postulated that decreased BDNF levels in neurodegenerative disorders leads to an imbalance of excitatory/inhibitory neurotransmission that may contribute to neuronal loss [85]. Thus, increased *Fxyd7* and *Fstl4* expression may be a compensatory response to maintain normal excitability in Purkinje neurons resistant to neurodegeneration [86]. *Them6* encodes a thioesterase of unknown function. These data on differential gene expression in *Npc1^−/−^* Purkinje neurons should be considered as hypothesis generating, and further work will be needed to determine if altered expression of any of these genes contributes directly to Purkinje neuron loss.

With respect to glial cells, oligodendrocytes and ependymal secretory cells had a relatively large number of transcriptomic changes in the *Npc1^−/−^* mice. Interestingly, oligodendrocytes were the only cell type in which we observed increased expression of *Npc2*. *Npc1^−/−^* oligodendrocytes also show increased expression of *Ptgds* (prostaglandin-H2 d-isomerase) and *Gstp1* (glutathione S-transferase P1) two genes where increased expression has been proposed to be neuroprotective [87,88]. In addition to immune-related (*B2m*, *C1qa*, *C1qb*, *C1qc*, *Lyz2*, *Tyrobp*) and lysosomal (*Ctsl*, *Ctss*, *Ctsz*, *Hexb*) genes, ependymal secretory cells express a number of genes that may be in response to the ongoing neurodegeneration. Increased expression of *Trem2*, *Manf*, and *Spp1* may be neuroprotective. *TREM2* variants are associated with Alzheimer disease [89] and soluble TREM2 (sTREM2) enhances microglial uptake and degradation of Amyloid beta [90]. Cerebrospinal fluid (CSF) levels of sTREM2 may be an early biomarker for onset of Alzheimer disease [91]. *Manf* encodes mesencephalic astrocyte-derived neurotrophic factor and may protect against Aβ toxicity [92]. *Spp1* encodes osteopontin, a cytokine that appears to have both neuroprotective and neurotoxic properties in neurodegenerative disorders [93]. Expression of solute carrier family 1 member 3 (*Slc1a3*) was utilized as a marker gene to identity astrocytes. When averaged across all cells, *Slc1a3* expression was lower in *Npc1^−/−^* astrocytes. Although not significant in our dataset, this observation is consistent with western blot data showing decreased protein expression [94]. *Npc1^−/−^* astrocytes have transcriptomes that represent a continuum between the homeostatic (*Slc1a3*^high^) and reactive (*Slc1a3*^low^) states. *Slc1a3* encodes the main ionotropic glutamate transporter in the cerebellum (GLAST, EAAT1). Glutamate is an excitatory neurotransmitter and elevated extracellular glutamate levels can be neurotoxic. Mutation of *SLC1A3* underlies type-6 episodic ataxia [95]. Thus, if decreased expression of *Slc1a3* leads to a functional deficiency of glutamate uptake by *Npc1^−/−^* astrocytes, then glutamate neurotoxicity may be a contributing factor in Purkinje neuron loss. Future work will focus on determining if these differentially expressed genes contribute to NPC1 neuropathology or if their products are potential biomarkers of NPC1 disease progression.

In order to obtain insight into pathogenic processes that occur prior to neuronal loss in the NPC1 mouse model, we also obtained single cell transcriptomic data on cerebellum isolated from asymptomatic three-week-old *Npc1^+/+^* and *Npc1^−/−^* mice. We did not anticipate the paucity of transcriptomic changes at this age in cells other than microglia. Central nervous system cholesterol storage is present at birth and increases markedly between postnatal day 1 and seven weeks of age occurs early in the disease process and increases [96]. Significant unesterified cholesterol storage is present in *Npc1^−/−^* microglia at birth [16]. Cholesterol storage in Purkinje neurons can be visualized in Purkinje neurons by postnatal day 9 [97] and Purkinje neuron loss is first observed around postnatal day 40 [98]. Sphingoid bases and gangliosides are elevated at 4 weeks of age and increase significantly between 4 and 7 weeks of age [99]. Given that significant unesterified cholesterol and lipid storage is present in asymptomatic *Npc1^−/−^* mice, we expected to observe early transcriptome changes that preceded loss of Purkinje neurons. It is possible that cell autonomous death of Purkinje neurons is driven by low abundance transcripts that are not detectable with the current experimental approach. Alternatively, although significant unesterified cholesterol accumulation is present at this age, the 3-week time point may precede significant impact on Purkinje neuron function. However, we did observe a few early transcriptome differences in vascular endothelial cells and multiple changes in microglia.

Four transcripts (*Apoe*, *Mt2*, *Timp3*, and *Slc16a1*) showed altered expression in three-week *Npc1^−/−^* endothelial cells. *Mt2* encodes metallothionein 2, and was also observed, along with *Mt1*, to also have increased expression in astrocytes from seven-week-old *Npc1^−/−^* mice. Metallothionines are cysteine rich proteins that are involved in the binding, transport and detoxification of heavy metals. Alterations of heavy metal homeostasis has been described in NPC1 [100]. These data suggest that this may be an early pathological component of NPC1. *Timp3* encodes tissue inhibitor of matrix Metalloproteinase-3, a soluble protein that promotes blood–brain barrier integrity and is neuroprotective in traumatic brain injury models [101]. *Slc16a1* encodes a proton-coupled transporter that transports monocarboxylates such as lactate, pyruvate and ketone bodies across the plasma membrane. This could reflect a switch toward glycolytic metabolism previously reported in NPC1 [16,46]. These data indicate that endothelial cells may be one of the first non-immune cell types to respond to initial NPC1 metabolic changes.

In stark contrast to the general lack of genotype related difference in neuronal transcriptomes, at three weeks of age, *Npc1^−/−^* microglia already show reduced expression of linage (*Tmem119*) and increased expression of activation (*Cd68*, *Igf1* and *Gapdh*) markers. These observations are consistent with our prior data showing corresponding changes at the protein level [27]. This observation of microglial activation preceding significant changes in neuronal transcriptome changes is consistent with the recent observation by Kavetsky et al. [102] that showed increased interactions and engulfment of dendrites by microglia in the molecular layer prior to Purkinje neuron degeneration. The *Npc1^−/−^* microglia transcriptomes from seven-week-old mice can be classified as disease-associated microglia initially described in Alzheimer disease [29]. Our current data support the concept that NPC1 microglia have both neuroprotective and neurotoxic activity [27]. This dual role will confound efforts to treat NPC1 by general approaches to reduce microglia activity. Our data also clearly show that microglial activation is an early, potentially neuroprotective, process that proceeds significant alterations in neuronal transcriptomes.

Although microglia are the endogenous immune cells in the central nervous system, our scRNAseq data show that multiple *Npc1^−/−^* cell types express genes related to the innate immune response by seven weeks of age. In fact, this innate immunity response predominates in detectable transcriptomic changes. It is possible that these cells are responding to a “danger signal” to which the microglia are more responsive. Given the mixed neuroprotective and neurotoxic components of neuroinflammation, it is possible that the early specific microglial response may be neuroprotective, but the change to a general chronic innate immune response by nonimmune cells may be neurotoxic. It is thus plausible that targeting the general nonimmune cell innate immune response may be a potential therapeutic target and potentially more effective than therapies directed at microglia alone.

To our knowledge this is the first report of scRNAseq comparing transcriptomes from NPC1 and control cerebellar tissue. These data complement prior studies comparing whole tissue transcriptomes. This work should be considered an exploratory, hypothesis generating study. Future work will be needed to establish if proteins corresponding transcriptome changes can be exploited as biomarkers to monitor disease progression and therapy, or if the proposed pathological mechanisms contribute to NPC1 cerebellar neuropathology and can be exploited as therapeutic targets.

## 4. Materials and Methods

### 4.1. Mouse Models and Phenotypic Evaluation

All mouse experiments were approved by the NICHD Animal Care and Use Committee Protocol #15-002 and #18-002. BALB/c-*Npc1*^+/−^ were obtained from the Jackson Laboratory (Bar Harbor, ME, USA). Heterozygous *Npc1^+/nih^* mice (BALB/cNctr-*Npc1^m1N^*/J strain) [103] were bred to obtain control (*Npc1^+/+^*) and mutant (*Npc1^−/−^*) littermates. Mice genotype were identified by PCR using the primers listed in Appendix A. Water and mouse chow, control or with 290 mg·kg^−1^ PLX3397 (Selleckchem, Houston, TX, USA), were available ad libitum. Genotyping PCR was performed using ear punch DNA. A humane survival endpoint was defined as hunched posture, reluctance to move, inability to remain upright when moving, and weight loss >30% of peak weight.

### 4.2. Generation of Single-Cell Suspensions, Single-Cell Reverse Transcription, Library Preparation, and Sequencing from Mouse Cerebella

Resected cerebella were dissociated using papain dissociation for 15 min according to the manufacturer’s instruction for 15 min (Worthington Biochemical Corporation, Lakewood, NJ, USA). Cell dissociation quality, density and viability were evaluated by trypan blue staining and TC20 automated cell counter (Bio-Rad, Hercules, CA, USA) and direct observation of the cell. Two pairs of three- and seven-week-old mouse cerebella (*Npc1*^+/+^ and *Npc1*^−/−^) and were run separately on chromium 10x Single Cell Chips (10x Genomics, San Francisco, CA, USA). Libraries were prepared using Chromium Single Cell Library kit V3 for the seven-week-old samples and V3.1 for the three-week-old samples (10x Genomics) and sequenced on an Illumina HiSeq2500 using 100bp paired-end sequencing. Raw sequencing data were processed using Cell Ranger (v1.3.1, 10x Genomics) to produce gene-level counts for each cell in each sample. Each sample was aggregated to form a single matrix. All subsequent analysis was performed using Automated Single Cell Analysis Platform VI [104]. Cells with greater than 95% of genes with zero assigned reads were removed. Genes with counts in less than four cells were filtered out. Remaining blood cells (Hbb^+^) seen in the final data were manually excluded from the analysis. The data were normalized using Voom [105]. Hierarchical clustering on the t-SNE dataset using the K-mean method were applied. The lists of differentially expressed genes, identified using limma [106], for each cluster were generated by testing for genes that had a log2 fold change greater than one between cells in one cluster compared with all other cells listed in Appendix A. Pathway analysis using gene ontology results are in Appendix A. Cell population with less than 10 events were not included in the comparisons. Raw data are available on SRA under the accession number: PRJNA606815.

### 4.3. Immunofluorescence and Flow Cytometry Analysis

Brain cerebellar tissue was processed as described. Mice were euthanized at seven weeks by CO_2_ asphyxiation and transcardially perfused with ice-cold 4% paraformaldehyde in PBS, pH 7.4. The brains were post fixed in 4% paraformaldehyde solution for 24 h and then cryoprotected in 30% sucrose (Sigma-Aldrich Millipore, St. Louis, MO, USA). Cerebellar tissues were cryostat-sectioned parasagittally (20 μm) and floating sections were collected in PBS with 0.25% Triton X-100 (Sigma-Aldrich Millipore, St. Louis, MO, USA). Sections were stained at 4 °C overnight with the antibody and dyes listed in Appendix A. Images were taken using a Zeiss Axio Observer Z1 microscope fitted with an automated scanning stage, Colibri II LED illumination, and Zeiss ZEN2 software using a high-res AxioCam MRm camera and a 20× objective (Zeiss, Oberkochen Germany). Each fluorophore channel was pseudo-colored in ZEN2, exported as .czi file, and analyzed using ImageJ [107]. Gating strategy and cell preparation for FACS analysis on mice cerebella are extensively described in [16]. Mean Fluorescence intensities were extracted using FlowJo software (TreeStar, Ashland, Covington, OR, USA).

### 4.4. Statistical and Informatic Analysis

All statistical comparisons outside of the scRNAseq analysis were performed with GraphPad Prism 5 software (San Diego, CA, USA). All box and whiskers plot display the Tukey method and have Mann-Whitney test *p* < 0.01 when comparing *Npc1^+/+^* to *Npc1^−/−^*. For the lifespan estimation, data were plotted in Kaplan–Meier survival curves, using the log rank test for survival analysis, plotting percent survival as a function of time in days. Results were considered significant if the *p* value was <0.05. Figures and Tables were created in Microsoft Office 365 or 2016 (Redmond, WA, USA).

## Figures and Tables

**Figure 1 ijms-21-05368-f001:**
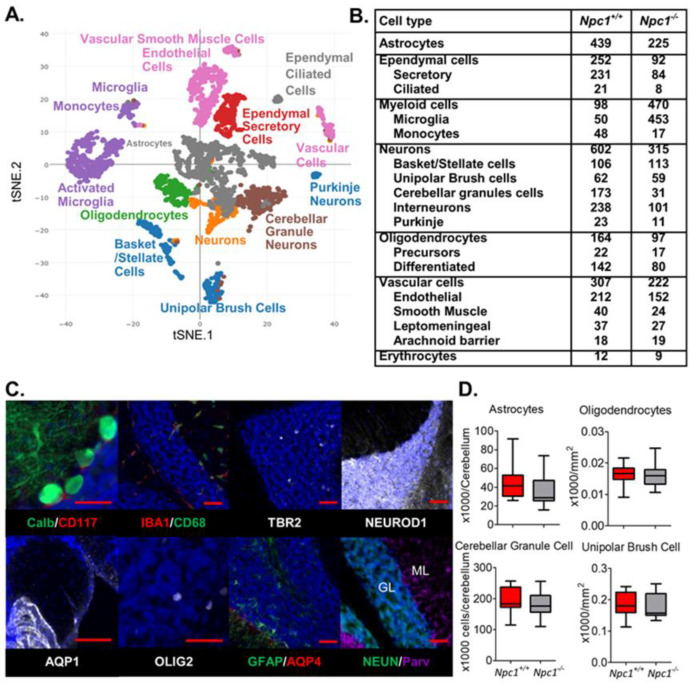
Identification of cerebellar cell populations. (**A**) Automatic K-means clustering was used to generate a t-SNE plot of single cell transcriptomes from *Npc1^+/+^* and *Npc1^−/−^* cerebellar tissue at seven weeks of age. (**B**) Number of cerebellar cell type specific transcriptomes obtained from *Npc1^+/+^* and *Npc1^−/−^* seven-week-old mice. (**C**) Representative immunostaining using antibodies corresponding to signature transcripts in seven-week-old mouse cerebella. Calb (Calbindin, Purkinje neurons); CD117 (basket/stellate neurons); IBA1 (microglia); CD68 (myeloid cells); TBR2 (unipolar brush neurons); NEUROD1 (cerebellar granule neurons); AQP1 (ependymal secretory cells); OLIG2 (oligodendrocytes); GFAP (astrocytes); AQP4 (astrocytes); NEUN (neurons); Parv (Parvalbumin, interneurons). Scale bar = 50 µm. (**D**) Histopathological quantification of number of cerebellar granule neurons, oligodendrocytes, astrocytes, and unipolar brush cells in parasagittal sections of seven-week-old cerebella. *N* > 6.

**Figure 2 ijms-21-05368-f002:**
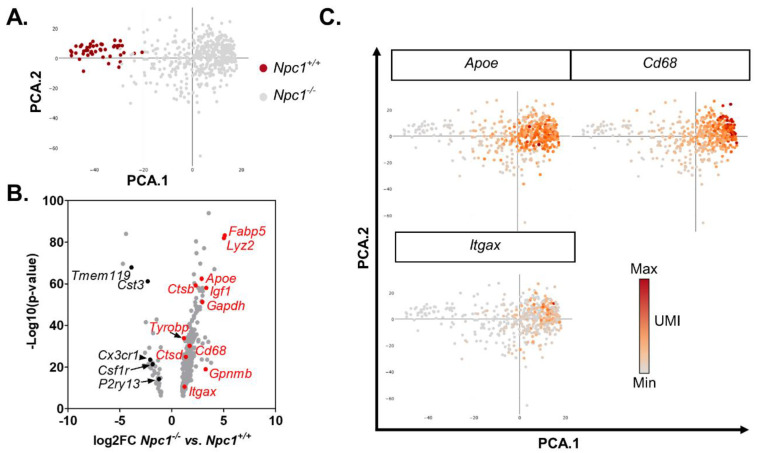
Differentially expressed genes in microglia. (**A**) Principal component analysis (PCA) of gene expression in microglia from seven-week-old *Npc1^+/+^* (red) and *Npc1^−/−^* (gray) cerebellar tissue. (**B**) Volcano plot of differential gene expression between control and *Npc1* mutant mice. (**C**) PCA plots showing expression level of microglial activation markers (*Apoe*, *Cd68*, and *Igax*). Increasing red intensity corresponds to increasing unique molecular identifier (UMI) counts.

**Figure 3 ijms-21-05368-f003:**
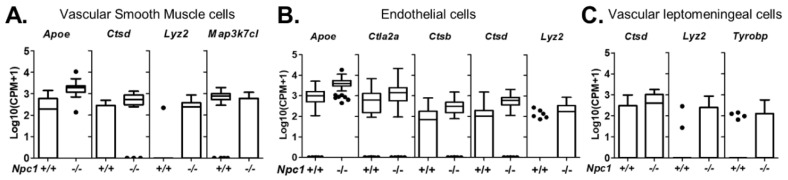
Differentially expressed genes in vascular cells. Tukey box plots of differentially expressed genes in vascular smooth muscle cells (**A**), endothelial cells (**B**), and vascular leptomeningeal cells (**C**).

**Figure 4 ijms-21-05368-f004:**
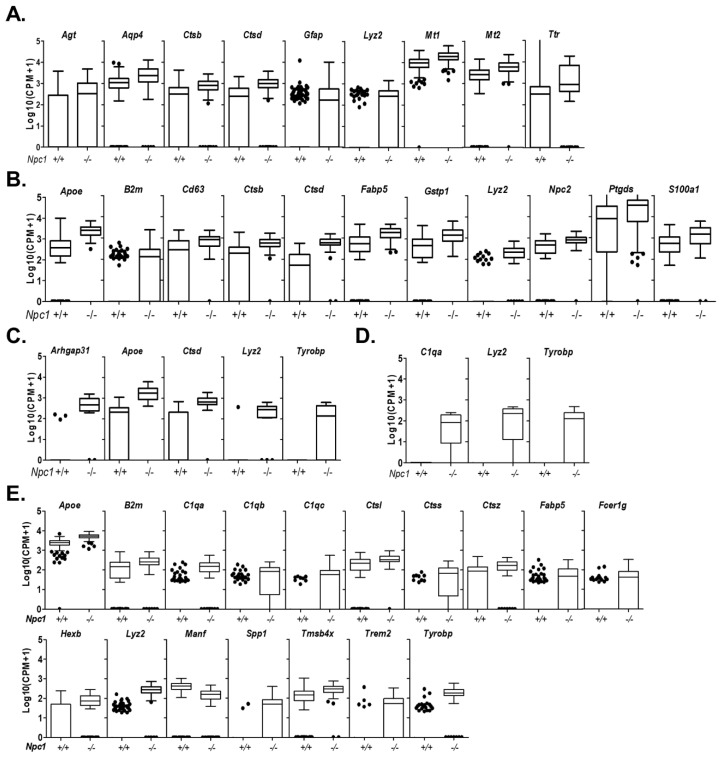
Differentially expressed genes in glial cells. Tukey box plot of differentially expressed genes in astrocytes (**A**), oligodendrocyte precursors (**B**), differentiated oligodendrocytes (**C**), ependymal ciliated (**D**), and ependymal secretory cells (**E**) between genotype.

**Figure 5 ijms-21-05368-f005:**
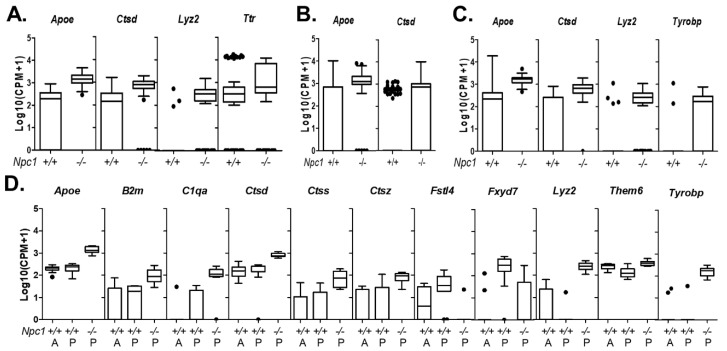
Differentially expressed genes in neuronal cells. Tukey box plots of differentially expressed genes in basket/stellate neurons (**A**), granule neurons (**B**), unipolar brush cells (**C**), and Purkinje neurons (**D**). In control cerebellum, anterior (**A**) and posterior (**B**) Purkinje neurons could be differentiated. In *Npc1^−/−^* cerebellum from seven-week-old mice, only posterior Purkinje neurons were identified.

**Figure 6 ijms-21-05368-f006:**
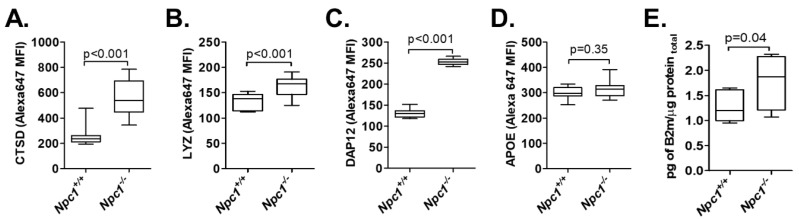
Protein validation of selected differentially expressed transcripts identified in 7-week old *Npc1^−/−^* cerebella. Immunohistochemistry quantification of cathepsin D (**A**), lysozyme (**B**), DAP12 (**C**), and apolipoprotein E (APOE) (**D**). Representative examples of cerebellar staining are provided in Appendix A. β2 microglobulin (**E**) was quantified by ELISA. *N* > 6.

**Figure 7 ijms-21-05368-f007:**
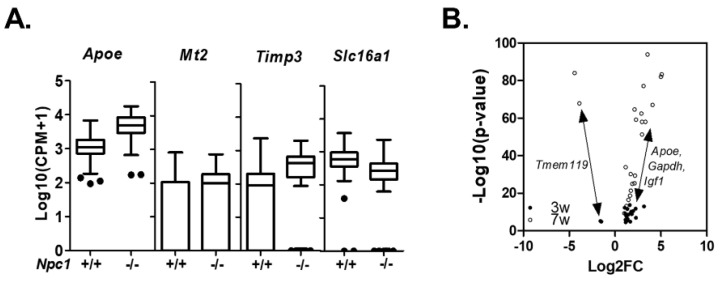
Differentially expressed genes at three weeks of age. (**A**) Tukey Box plot of genes showing differential expression *Npc1+/+* and *Npc1^−/−^* endothelial cells. (**B**) Volcano plot of significantly differentially expressed genes in microglia from *Npc1^+/+^* and *Npc1^−/−^.* X axis Log2 fold change between *Npc1^−/−^* and *Npc1^+/+^* microglia. Y axis is the -Log10 *p*-value. Black dots and clear circles indicate data from three- and seven-week-old mice, respectively.

**Figure 8 ijms-21-05368-f008:**
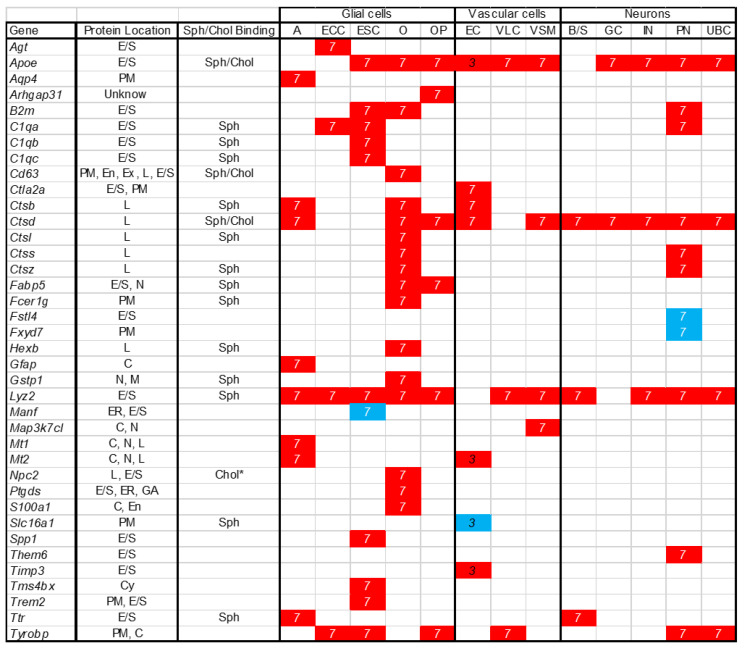
Single cell expression data for 38 differentially expressed gene. Increased (red) and decreased (blue) at both three and seven weeks is indicated for specific cell types. A: astrocytes; ECC: ependymal ciliated cells; ESC: ependymal secretory cells; O: oligodendrocytes; OP: oligodendrocytes precursor; EC: endothelial cells; VLC: vascular leptomeningeal cells; VSM: vascular smooth muscle cells; B/S: basket/stellate cells; GC: granule cells; IN: interneurons; PN: Purkinje neurons; UBC: unipolar brush cells. Cellular location of the encoded proteins is from Uniprot. E/S: extracellular space or secreted; PM: plasma membrane; C: cytoplasm; N: nucleus; ER: endoplasmic reticulum; GA: Golgi apparatus; L: lysosome; En: endosome; Ex: exosome; M: mitochondrion; Cy: cytoskeleton. Sphingolipid binding is from Haberkant et al. [57]. Cholesterol binding is from Hulce et al. [63] and * [64].

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
