# Peer review of "Single Cell Transcriptome Analysis of Niemann–Pick Disease, Type C1 Cerebella"

_ijms, 2020, doi:10.3390/ijms21155368_

Round 1
Reviewer 1 Report
A very nice piece of work - I have some minor comments:
Since relatively few proteins that have altered expression in the cerebellum of NPC mice I think that a table summarising the results should be prominently displayed in the results including subcellular location and function (where known).
Discussion of these results would be enhanced if they could describe how lysosomal storage may lead to changes in expression and more generally what feedback mechanisms are thought to operate between protein levels and mRNA control.
Any differences of storage of lipids between 3 and 7wk old mice should be discussed.
It would be useful to compare the proteins that they find with lipid binding proteins identified by lipid probes such as those used by cravatt for cholesterol and haberkant for sphingolipids I believe both find cathepsins in their screens as well as solute carriers. Although these studies didin't use purkinje cells are there any closely related proteins in common? Are any of them present at membrane contact sites?
The abbreviations listed are incomplete
Author Response
A very nice piece of work - I have some minor comments:
Thank you
Since relatively few proteins that have altered expression in the cerebellum of NPC mice I think that a table summarising the results should be prominently displayed in the results including subcellular location and function (where known).
We have added figure 8 to the manuscript to provide a prominent summary (page 17). Figure 8 provides the following information on the 38 differentially expressed genes:
- Increased or decreased expression at either 3 or 7 weeks
- Protein location
- Sphingosine or cholesterol binding (see below)
Even though the number of genes is limited, inclusion of functional information makes for a multipage table. Thus, we have left this information in Table S1 and provided additional information in the new figure 8.
Discussion of these results would be enhanced if they could describe how lysosomal storage may lead to changes in expression and more generally what feedback mechanisms are thought to operate between protein levels and mRNA control.
Because a significant portion of the differentially expressed genes encode lysosomal or excreted proteins, we now discuss the possibility that lysosomal exocytosis and loss of these proteins may result in feedback regulation of RNA expression (lines 446-453).
Any differences of storage of lipids between 3 and 7wk old mice should be discussed.
We have added a discussion about the age dependent increase of unesterified cholesterol and lipids. (lines 559-566).
It would be useful to compare the proteins that they find with lipid binding proteins identified by lipid probes such as those used by cravatt for cholesterol and haberkant for sphingolipids I believe both find cathepsins in their screens as well as solute carriers. Although these studies didin't use purkinje cells are there any closely related proteins in common? Are any of them present at membrane contact sites?
We would like to thank the reviewer for suggesting this comparison. Of the 38 differentially expressed genes identified in our study, 16 (42%) were identified by Haberkamp et al. as sphingolipid-binding proteins. In contrast only 3 were identified by cholesterol binding. We now include this observation in the discussion (lines 454-463) and figure 8. We assume that the reviewer is asking about Lysosomal/ER contact sites. We are not aware of any of the 38 proteins participating in the protein complexes at those contact points.
The abbreviations listed are incomplete
Thank you. This has now been corrected.
Reviewer 2 Report
This paper was described about single cell transcriptome analysis for cerebellar tissue of Niemann-Pick type C1. The authors showed increased expression of innate immune genes in every cell type including neuron and glia, and Disease Associate Microglia activation similar to Alzheimer disease. They also demonstrated that microglia activation precedes neuronal dysfunction. These observations provide valuable insight to the pathogenesis of NPC. I think this paper is well written and acceptable for publication in this journal.
Author Response
We appreciate these comments and would like to thank this reviewer for their time and effort.